# GeoDream: Disentangling 2D and Geometric Priors for High-Fidelity and Consistent 3D Generation

## Abstract

Text-to-3D generation by distilling pretrained large-scale text-to-image diffusion models has shown great promise but still suffers from inconsistent 3D geometric structures (Janus problems) and severe artifacts. The aforementioned problems mainly stem from 2D diffusion models lacking 3D awareness during the lifting. In this work, we present GeoDream, a novel method that incorporates explicit generalized 3D priors with 2D diffusion priors to enhance the capability of obtaining unambiguous 3D consistent geometric structures without sacrificing diversity or fidelity. Specifically, we first utilize a multi-view diffusion model to generate posed images and then construct cost volume from the predicted image, which serves as native **3D geometric priors**, ensuring spatial consistency in 3D space. Subsequently, we further propose to harness 3D geometric priors to unlock the great potential of 3D awareness in 2D diffusion priors via a disentangled design. Notably, disentangling 2D and 3D priors allows us to refine 3D geometric priors further. We justify that the refined 3D geometric priors aid in the 3D-aware capability of 2D diffusion priors, which in turn provides superior guidance for the refinement of 3D geometric priors. Our numerical and visual comparisons demonstrate that GeoDream generates more 3D consistent textured meshes with high-resolution realistic renderings (i.e., $1024 \times 1024$) and adheres more closely to semantic coherence.

Diffusion models Saharia et al. (2022); Rombach et al. (2022); Ramesh et al. (2022) have significantly advanced text-to-image synthesis. Inspired by their success, it is appealing to lift this success from 2D to 3D because this achievement holds significant potential impacts on the modern game and media industry. Template-based generators Chen et al. (2023a) and 3D native generative models Li et al. (2023b); Wang et al. (2023c); Mo et al. (2023); Nichol et al. (2022); Jun & Nichol (2023) provide a natural and direct approach to the lift. However, these methods usually show compelling results for limited categories due to the lack of extensive 3D data. Recently, the Score Distillation Sampling (SDS) Poole et al. (2022) and Variational Score Distillation (VSD) Wang et al. (2023d) have been introduced to optimize 3D representations such that images rendered from any viewpoints match the text-conditioned image distribution evaluated by a pretrained text-to-image (T2I) model. This is an exciting direction because it allows for generating 3D assets from any given text prompt, circumventing the need for any 3D data. Despite these methods yielding satisfactory results on a wide range of geometrically symmetrical 3D shapes, empirical observations indicate that SDS and VSD losses still suffer from inconsistent 3D geometric structures (Janus problems) Wikipedia (2023) and severe artifacts Wang et al. (2023d); Shi et al. (2023b) with asymmetric geometry. This is primarily due to the lack of 3D awareness in 2D diffusion models, which inherently makes the lifting from 2D observations into 3D ambiguous.

As a remedy, learning 3D priors from 3D datasets seems theoretically reasonable and correct. However, 3D data remains expensive and sparse compared to the plentifully available images. Therefore, the most promising avenue Qian et al. (2023); Shi et al. (2023b); Sun et al. (2023) presently is to equip 2D diffusion priors with 3D priors learned from relatively limited 3D data, aiming to achieve the best of both worlds. Recently, with the release of large-scale 3D datasets, Objaverse Deitke et al. (2023b) and Objaverse-XL Deitke et al. (2023a), a few works Liu et al. (2023c); Li et al. (2023c); Shi et al. (2023b); Ye et al. (2023) have attempted to finetune pre-trained 2D diffusion models using multi-view images rendered from 3D dataset. This involves obtaining multi-view images from the

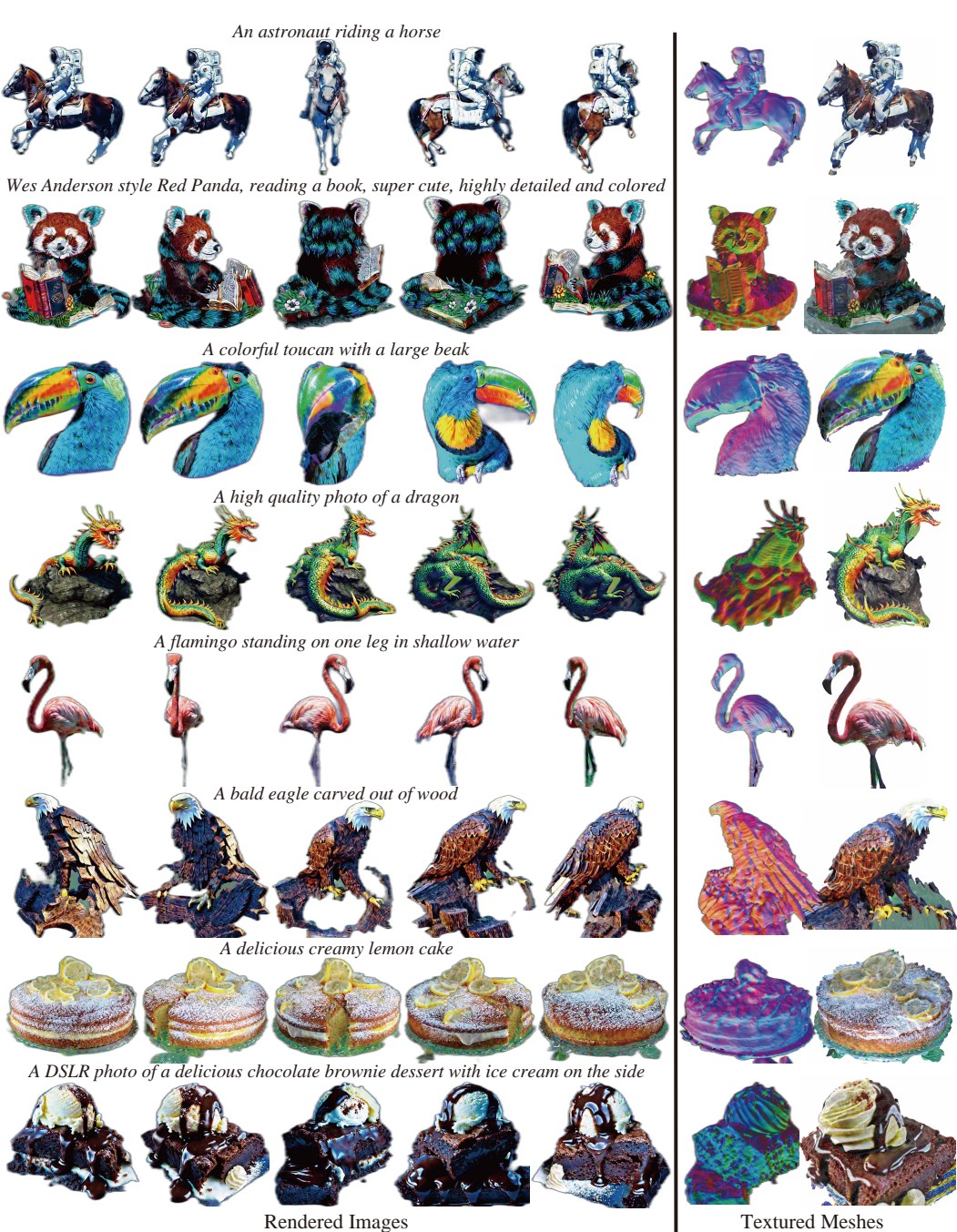

Figure 1: GeoDream alleviates the Janus problems by incorporating explicit 3D priors with 2D diffusion priors. GeoDream generates consistent multi-view rendered images and rich details textured meshes. We remove the rendering background to achieve a clearer visualization.

fine-tuned diffusion model conditioned on camera parameters and utilizing the clues of predicted multi-view consistency to infer 3D information. Nevertheless, these methods rely heavily on the consistency of content predicted across different source views. Such inconsistencies between the predicted multiple views become particularly noticeable, especially in imaginative and uncommon cases beyond the training data distribution, resulting in over-smoothing and the loss of semantic geometries in the generated 3D assets.

To resolve this issue, we introduce GeoDream, a novel method that incorporates explicit generalized 3D priors with 2D diffusion priors to enhance the capability of obtaining unambiguous 3D consistent geometric structures, while maintaining diversity and high fidelity. Our contributions are listed below. **(i)** In contrast to methods mentioned above that rely on consistency between multi-view priors, we propose to obtain 3D native priors in the 3D world space, which are well-suited to handle the inherent lack of perfect consistency within the multi-view predicted priors, and naturally free from inconsistencies caused by camera viewpoint transition. **(ii)** We justify that disentangling 3D and 2D priors is a potentially exciting direction for maintaining both the generalization of 2D diffusion priors and the consistency of 3D priors. In other words, providing hints through 3D priors to unlock the great potential of 3D awareness in 2D diffusion priors, without the need for invasive finetune 2D diffusion models.

Specifically, we start by reconstructing cost volume as native 3D priors by aggregating the predicted multi-view 2D images into 3D space. Such aggregation operations have been widely used in MVS-based techniques Yao et al. (2018); Zhang et al. (2022); Long et al. (2022); Liu et al. (2023b), which are known to be robust and generalized to provide valuable cues for geometric reasoning. We find that such operations are well-suited for handling imperfect and inconsistent multi-view predictions. The reason is that they involve multi-view information aggregation, which helps filter out inconsistent content to some extent, rather than dealing with each view individually. Foremost, we conduct extensive experiments to demonstrate that our proposed 3D priors adapt to multiple views predicted by various off-the-shelf multi-view diffusion models, such as Zero123 Liu et al. (2023c), MVDream Shi et al. (2023b) and Zero123++ Shi et al. (2023a). Moreover, we introduce a critical viewpoint sampling strategy to promote the stability of the 3D priors. Despite recent attempts to reconstruct cost volume as 3D priors, such as One2345 Liu et al. (2023b) and SyncDreamer Liu et al. (2023d). However, they treat the cost volume as a fixed 3D prior. In contrast, we propose a novel method that allows the integration of 2D diffusion priors and 3D priors in an optimizable way. We conduct extensive experiments to investigate the respective roles of 2D diffusion priors and 3D priors in 3D generation tasks. We hope that the incorporating design will bridge the gap between 2D and 3D priors, contributing to a harmonious synthesis of both.

Specifically, we further propose incorporating 3D priors with 2D diffusion priors in a disentangled solution. Existing multi-view diffusion priors are equipped with 2D diffusion priors in a coupled way, including generating multiple views as supervision Liu et al. (2023c); Shi et al. (2023a) or distilling the probability density as a loss Shi et al. (2023b); Li et al. (2023c); Sun et al. (2023); Qian et al. (2023) to compute gradients for optimizing 3D representations. Instead, we justify that leveraging the geometric clues provided by 3D priors can effectively unleash the great potential 3D awareness capability inherent in 2D diffusion priors, referred to as "disentangled design". Very recent works have started to explore how to evoke 3D-aware ability in 2D diffusion by altering score functions Hong et al. (2023) or negative text prompts Armandpour et al. (2023). These efforts have made surprising progress, yet the performance remains unstable regarding 3D consistency. Our insight is that going through geometric priors to unlock the great potential of 3D awareness in 2D diffusion is a promising direction that is both general and stable. Moreover, we rely solely on the awakened 3D-aware capability of 2D priors to guide the optimization of Neural Implicit Surfaces (NeuS) Wang et al. (2021) without the supervision of 3D priors, thereby avoiding compromising the inherent advantages of 2D priors in terms of generalization and creativity. We show that 3D priors can be further refined to boost rendering quality and geometric accuracy. The 2D diffusion priors benefit from gradually evolved 3D priors, which in turn provide superior guidance for unleashing the 2D priors. Finally, we use DMTet Shen et al. (2021); Lin et al. (2023) to extract textured mesh from optimized NeuS for mesh fine-tuning. Unlike previous work Poole et al. (2022); Wang et al. (2023d); Lin et al. (2023) attempt to increase the rendering resolution, which typically suffers from over-saturation issues, we successfully increase the rendering resolution from 512 to 1024. We hypothesize that the improved results are aided and abetted by 3D priors that provide more plausible geometry and realistic texture, making the optimization easier, because the rendered image is closer

Table 1: Comparison of design space.

| Method | One2345 | MVDream | GSGEN | Ours |
|---|---|---|---|---|
| Repr. | NeuS | NeRF | Gaussian | NeuS+DMTet |
| Resolution | 512 | 512 | 512 | **1024** |
| 3D guidance | Cost volume | Multi-Views | Point-E | Cost volume |
| 3D&2D | Only 3D priors | Entangled | Entangled | **Disentangled** |
| 3D priors | Fixed | Fixed | Fixed | **Optimizable** |

to diffused distributions. To comprehensively evaluate semantic coherence, to our knowledge, we are the first to propose $\mathrm{Uni3D_{score}}$ metric, lifting the measurement from 2D to 3D.

As summarized in Tab.1, we compared the latest methodsLiu et al. (2023b); Shi et al. (2023b); Chen et al. (2023b) in design space, including 3D representation, rendering resolution, forms of 3D guidance, the disentangling of 3D and 2D priors and the optimizability of 3D priors. As shown in Fig.1, GeoDream can yield $1024 \times 1024$ high-resolution rendered images and high-fidelity textured meshes while greatly alleviating the notorious Janus problems. In Sec.3.1, we conduct comprehensive evaluations that demonstrate the superiority of the 3D assets generated by GeoDream in terms of plausible geometry and delicate rendering details in visual appearance. To facilitate future research, we will release all the source code and test prompts.

## 1 RELATED WORK

**3D Generation Guided by 2D Priors.** Deep generative models have driven the field of 3D generation. Some efforts utilize Variational Auto Encoders (VAEs) Kingma & Welling (2013) for texture generation Henderson & Ferrari (2020); Henderson et al. (2020), while Generative Adversarial (GAN) Models Goodfellow et al. (2014) investigate 3D-aware GAN training Chan et al. (2022); Deng et al. (2022). Thus far, diffusion models have exhibited relatively better generalizability and training stability for diverse object generation compared to GANs and VAEs, and thus have gradually become recent focal points in 3D generation. Specifically, recent endeavors attempt to leverage the potent 2D diffusion priors to aid 3D generation by coupling it with a 3D representation, such as NeRF Mildenhall et al. (2021), DMTNet Shen et al. (2021), or NeuS Wang et al. (2021), among others, bypasses the necessity for extensive text-3D datasets for training 3D generative models. Such methods involve various techniques, including score distillation sampling schedules like SDS Wang et al. (2023a), SJC Poole et al. (2022), VSD Wang et al. (2023d) and ISM Liang et al. (2023) losses, which optimize the 3D representation by enhancing high likelihood evaluated by the 2D diffusion models. A coarse-to-fine training strategy Chen et al. (2023a) strengthens texture representation, decoupling geometric and texture aspects of 3D representation for finer optimization Lin et al. (2023), improving 3D representation Tang et al. (2023); Chen et al. (2023b). Although these methods demonstrate the ability to generate photo-realistic and diverse 3D assets with user-provided textual prompts, they are prone to the notorious 3D inconsistency issues (Janus problems) during the lifting process due to their reliance on 2D diffusion models for training, which lack 3D knowledge. Despite some current methods attempting to address 3D inconsistency by altering score functions Hong et al. (2023) or negative text prompts Armandpour et al. (2023), performance remains instability in terms of 3D consistency. In this work, we aim to explore the distinctive advantages of incorporating explicit 3D priors with 2D priors, enabling the generation of highly detailed 3D objects while remarkably mitigating 3D inconsistency issues.

**3D Generation Guided by 3D Priors.** Learning 3D priors from 3D datasets seems theoretically reasonable and correct for enhancing the coherency of 3D generation Liu et al. (2023c;b); Lin et al. (2023); Melas-Kyriazi et al. (2023); Xu et al. (2023a); Purushwalkam & Naik (2023). Therefore, various 3D latent diffusion models trained on 3D data have been recently introduced, including those using Tri-plane Shue et al. (2023) or feature grid Wang et al. (2023b); Liu et al. (2023d) encoding 3D representations into the latent space. Additionally, OpenAI has explored models aiming to directly generate 3D formats using several million internal 3D shapes, such as point clouds Nichol et al. (2022) or neural radiance fields Jun & Nichol (2023). However, their generalizability to the scope of their 2D counterparts remains unverified, due to the relative sparsity of 3D data compared to the abundance of available 2D images. Consequently, the most promising avenue currently is to equip 2D diffusion priors with 3D priors learned from relatively limited 3D data, intending to achieve the best of both worlds. Recently, with the release of a large-scale 3D dataset called Objaverse Deitke et al. (2023b) and Objaverse-XL Deitke et al. (2023a), some work Liu et al. (2023c); Yang et al. (2023); Liu et al. (2023b); Li et al. (2023c); Shi et al. (2023b); Ye et al. (2023); Cao et al. (2023); Xu et al. (2023b); Li et al. (2023a); Liu et al. (2023a) has attempted to fine-tune pre-trained 2D diffusion models using multi-view images rendered from 3D data. This aims to generate multi-view images from the fine-tuned diffusion model conditioned on camera parameters and utilize the

Figure 2: The overview of GeoDream. (a) 3D priors training. (b) Incorporating 3D priors with 2D diffusion priors.

clues of predicted multi-view consistency to assist in inferring 3D information. Nevertheless, these methods heavily depend on the absolute consistency of content predicted across different views. Nonetheless, their efforts to utilize 3D self-attention Shi et al. (2023b); Yang et al. (2023) for feature exchange between different views, to correlate multi-view features using 3D-aware attention Ye et al. (2023), to transform RGB predictions into coarser Canonical Coordinates Map predictions Li et al. (2023c), to transform RGB predictions into normal-depth predictions Liu et al. (2023e); Qiu et al. (2023) for mitigate the negative impact of inconsistencies. The performance of such methods frequently exacerbates inconsistencies and unrealistic rendering quality in uncommon cases, due to the absence of explicit constraints between different predicted viewpoints within 3D space. In this work, we incorporate explicit generalized 3D priors into 2D diffusion priors. These explicit 3D priors fundamentally ensure consistency in 3D space and avoid the independence of multi-view priors across source views.

## 2 METHOD

We focus on generating 3D content with consistently accurate geometry and delicate visual detail, by equipping 2D diffusion priors with the capability to produce 3D consistent geometry while re-taining their generalizability. The overview of GeoDream is shown in Fig.2. GeoDream consists of the following two stages. i) During 3D priors training, we build upon the One-2-3-45 Liu et al. (2023b), which encodes geometry into cost volume $V$ and geometry MLP decoder $f_g$. In addition, the appearance of the object is modeled to cost volume $V$ and texture MLP decoder $f_t$. We refer to the trained geometric decoder $f_g$ and appearance decoder $f_t$ with cost volume $V$ as native 3D geometric priors and appearance priors, as shown in Fig.2 (a). Details in Sec.2.1. ii) During priors refinement, we show that geometric priors can be further fine-tuned to boost rendering quality and geometric accuracy by combining a 2D diffusion model, as shown in Fig.2 (b). Detials in Sec.2.2.

### 2.1 GENERALIZABLE 3D PRIORS TRAINING

We start by reconstructing cost volume $V$ as native 3D priors by aggregating the 2D image features into 3D space, which provides valuable cues for geometric reasoning in the priors refinement stage.

**Cost Volume Construction.** Following MVS-based methods Yao et al. (2018); Zhang et al. (2022); Long et al. (2022); Liu et al. (2023b), given multi-view images $I = \{(I_i)_{i=0}^{N-1}\}$, we extract 2D feature maps $F = \{(F_i)_{i=0}^{N-1}\}$ using a 2D feature network $f_{2D}$. The volume reconstruction model takes posed 2D feature maps $F$ as input and outputs cost volume $V$ with per-voxel neural features in voxels. Specifically, for each voxel centered at 3D location $h$, the per-voxel neural feature is computed by projecting each location $h$ to $N$ image feature planes and then fetching the variance of the features at the location of the projection. We use Var to denote the variance operation and $P$ to denote the projection procedure. We then use a sparse 3D CNN $f_{3D}$ to process the variance features per voxel to regress the cost volume, as formulated by,

$$V = f_{3D}( \text{Var}\{P(F_i, h)\}_{i=0}^{N-1}),$$

(1)

where the variance operation is invariant to the number $N$ of input images. We find that such an operation is well-suited for handling imperfect and inconsistent multi-view predictions, due to involving information aggregation rather than dealing with each view individually.

**Geometry and Texture Decoder.** The cost volume $V$ is directly decoded into signed distance function values (SDF) and color information using the corresponding geometry MLP decoder $f_g$ and texture MLP decoder $f_t$. For any arbitrary query point $x \in \mathbb{R}^3$, we get the SDF $s$ and color $c$ as

$$s(x) = f_g(E(x), V(x)), \tag{2}$$

$$c(x) = f_t(\{P(F_i, x)\}_{i=0}^{N-1}, V(x), \{\Delta d_i\}_{i=0}^{N-1}), \tag{3}$$

where $E$ denotes position encoding, $V(p)$ denotes tri-linearly interpolated feature from cost volume at query point $x$, $\Delta d_i = d - d_i$ is the viewing direction of the query ray relative to the viewing direction of the $ith$ multi-view image.

The final rendered image $I'$ is achieved via SDF-based differentiable volume rendering. In this work, we get the pre-trained parameters of the $f_g$, $f_t$, and $f_{3D}$ networks from the One-2-3-45 Liu et al. (2023b), which is trained on ground truth images $I$ rendered from the Objaverse dataset with a loss

$$\mathcal{L}_{rgb} = ||I - I'||_2, \tag{4}$$

where $I' = R(\{s(x_j), c(x_j))\}_{j=0}^{M-1})$, $M$ denotes sampling $M$ query points along the ray of viewing direction and $R$ denotes volume rendering.

## 2.2 Priors Refinement

We present how to further finetune the geometric priors obtained from 3D priors training stage, i.e., optimizable cost volume $V$ and the fixed pre-trained geometric decoder $f_g$, using the 2D diffusion priors, as shown in Fig.2 (b). During priors refinement stage, we replace the $N$ ground truth rendered images with multi-view diffusion model predictions. In contrast to One-2-3-45, GeoDream is not limited to the Zero123 Liu et al. (2023c) predictions. We conduct extensive experiments with various multi-view diffusion models, such as MVDream Shi et al. (2023b) and Zero123++ Shi et al. (2023a). We also introduce a critical viewpoint sampling strategy to ensure GeoDream robustly adapts to various multi-view diffusion models, rather than being limited to just one. Overall, we justify that by decoupling 3D and 2D diffusion priors, GeoDream unlocks the immense potential of 3D awareness in the 2D diffusion model, avoiding the tendency to produce canonical views, resulting in 3D assets featuring multiple faces and collapsed geometry. Thanks to the decoupling, GeoDream maintains the generalization and imaginativeness of 2D diffusion priors, while also exploring the significant role that geometric priors play in improving appearance modeling.

**Multi-View Images Generation.** The rapid advancement of 3D generation has provided a wide range of methods available for generating multi-view images, such as Zero123 Liu et al. (2023c), MVDream Shi et al. (2023b), and Zero123++ Shi et al. (2023a). Given a set of predefined camera poses $\{(R_i, T_i)_{i=0}^{N-1}\}$ and a user-provided condition $c$, we utilize a fixed multi-view diffusion $f_{mv}$ to predict posed images $I_p = \{(I_i^p)_{i=0}^{N-1}\}$ and extract 2D feature maps $F_p = \{(F_i^p)_{i=0}^{N-1}\}$,

$$F_i^p = f_{2D}(f_{mv}(c, R_i, T_i)), \tag{5}$$

where $R \in \mathbb{R}^{3\times3}$, $T \in \mathbb{R}^{3\times3}$ respectively denote relative camera rotation and translation of the default viewpoint.

**3D Geometric Priors.** By replacing $F_i$ in Eq.1 into $F_i^p$, we obtain the value of SDF at an arbitrary query point $x$ defined in Eq.2,

$$V_p = f_{3D}(\text{Var}\{P(F_i^p, h)\}_{i=0}^{N-1}), \tag{6}$$

$$s_p(x) = f_g(E(x), V_p(x)), \tag{7}$$

where $s_p(x)$ is treated as a geometric prior since it encodes the hidden geometric clues in the predicted multiple views.

**Texture Decoder.** We propose to drop the pre-trained texture priors $f_t$ defined in Eq.3 because we empirically find that texture priors tend to generate 3D assets with lighting and texture styles similar to the rendered dataset. We choose Instant NGP Müller et al. (2022) for efficient high-resolution

texture encoding. Specifically, for any arbitrary query point $x \in \mathbb{R}^3$, a learnable hash encoding $h_\Omega$ is decoded into a color $c$ using initialized texture decoder $f'_t$, as formulated by,

$$c_p(x) = f'_t(h_\Omega(x), x), \tag{8}$$

where $h_\Omega(x)$ denotes the looked-up feature vector from $h_\Omega$ at query point $x$.

**Texture and Geometry Refinement.** To incorporate 3D geometric priors with 2D diffusion priors, we minimize the VSD loss introduced in ProlificDreamer Wang et al. (2023d) to optimize the parameters of $\theta_1$ in cost volume $V$, $\theta_2$ in hash encoding $h_\Omega$ and $\theta_3$ in texture decoder $f'_t$. At each iteration, we sample a camera pose $o$ from a pre-defined distribution. We render 2D image $\hat{x}$ at pose $o$ by combining Eq.7 and Eq.8 via differential rendering $R$. Our objective function during priors refinement is to minimize the VSD loss $\mathcal{L}_{VSD}$, the corresponding gradient $\nabla_{\theta_1, \theta_2, \theta_3} \mathcal{L}_{VSD}$ is

$$\mathrm{E}_{t,\epsilon,o}[w(t)(\epsilon_{pretrain}(\hat{x}_t, t, c) - \epsilon_l(\hat{x}_t, t, c, o)) \frac{\partial \hat{x}}{\partial(\theta_1, \theta_2, \theta_3)}], \tag{9}$$

where $\hat{x}_t$ denotes a noisy rendered image in timestep $t$, $w(t)$ denotes a weighting function, $\epsilon_{pretrain}$ is a 2D pretrained diffusion model and $\epsilon_l$ is a trainable LoRA Hu et al. (2021) diffusion model with parameters of $l$. We propose to fix the geometry decoder $f_g$ conjointly with a learning rate decay strategy for the cost volume, aiming to maintain geometric priori cues as well as tuning to achieve better details in the early stage of optimization. More details on viewpoint sampling and learning rate decay strategy are provided in Sec.3.2.

**Mesh Fine-tuning.** For high-resolution rendering, we use DMTet Shen et al. (2021); Lin et al. (2023) to extract textured 3D mesh representation from optimized NeuS Wang et al. (2021). By minimizing the loss in Eq.9, we follow ProlificDreamer Wang et al. (2023d) first to optimize the geometry using the normal map and then optimize the texture. We empirically find that we can increase the rendering resolution from 512 to 1024. But unlike previous work Poole et al. (2022); Wang et al. (2023d); Lin et al. (2023), attempting to increase the rendering resolution suffers from over-saturation issues. We successfully increase the rendering resolution from 512 to 1024. We hypothesize that well-optimized results are aided and abetted by 3D priors that provide more plausible geometry and realistic texture, making the optimization easier, because the rendered image $\hat{x}$ is closer to diffused distributions at each iteration.

## 3 EXPERIMENT

### 3.1 RESULTS OF GEODREAM

**Baselines.** We report our performance with the latest 3D generation methods, including Dream-Fusion Poole et al. (2022), ProlificDreamer Wang et al. (2023d), MVDream Shi et al. (2023b), GSGEN Chen et al. (2023b), Fantasia3D Chen et al. (2023a), Magic123 Qian et al. (2023) and Wonder3D Long et al. (2023). Specifically, DreamFusion Poole et al. (2022), Fantasia3D Chen et al. (2023a) and ProlificDreamer Wang et al. (2023d) adopt a similar approach to optimize 3D representation through the score function of a 2D diffusion model, without intervening in 3D priors. We compare our results with these three methods, highlighting the distinct advantages of inferring 3D-consistent geometry and reducing artifacts by incorporating explicit 3D priors. Meanwhile, MVDream Shi et al. (2023b) and Wonder3D Long et al. (2023) are very recent proposals to use multi-view consistency priors, which are derived from finetuned multi-view diffusion models trained on synthetic multi-view rendering image data. GSGEN Chen et al. (2023b), on the other hand, addresses 3D inconsistency by initializing geometry with Point-E Nichol et al. (2022) generated shapes. By comparing these methods, we demonstrate that our introduced 3D priors offer greater generality in challenging and uncommon cases and effectively prevent the generation of 3D assets with lighting and texture styles similar to the synthetic rendered dataset. Magic123 Qian et al. (2023) adopts a coupled approach, optimizing the 3D representation by using both 3D and 2D priors as losses. The comparisons with Magic123 justify that the disentangling 3D and 2D priors allows for the simultaneous harnessing of the generalization capabilities of 2D diffusion priors and the 3D consistency of 3D priors. In contrast, Magic123 requires careful design of the balance weights between 3D and 2D loss to avoid compromising between the two types of priors.

**Implementations.** For DreamFusion Poole et al. (2022), ProlificDreamer Wang et al. (2023d) and Fantasia3D Chen et al. (2023a), we utilize their implementations in the ThreeStudio Guo et al. (2023) library for comparison. For MVDream Shi et al. (2023b), GSGEN Chen et al. (2023b), Magic123 Qian et al. (2023) and Wonder3D Long et al. (2023), we use their official implementation.

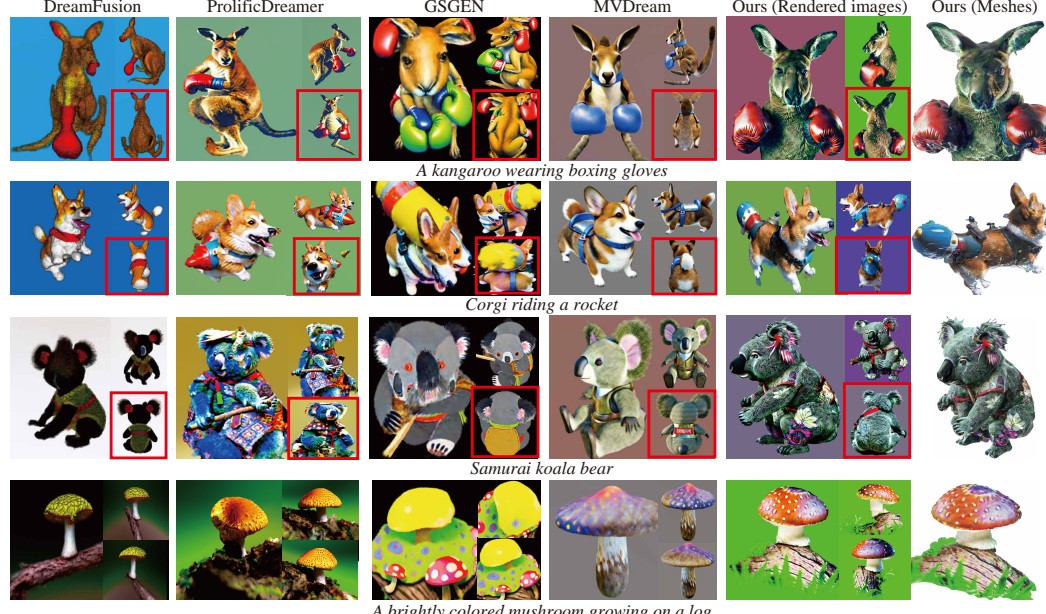

DreamFusion    ProlificDreamer    GSGEN    MVDream    Ours (Rendered images)    Ours (Meshes)

*A kangaroo wearing boxing gloves*

*Corgi riding a rocket*

*Samurai koala bear*

*A brightly colored mushroom growing on a log*

Figure 3: Qualitative comparison with baselines. Back views are highlighted with red rectangles for distinct observation of multiple faces.

**Experiment Setup.** We collected 35 prompts from various sources, including prompts from previous work Shi et al. (2023b); Long et al. (2023) and real user inputs in the wild. To comprehensively assess 3D consistency and semantic coherence, we intentionally selected more prompts indicating asymmetric geometric structures (80% of the collected prompts) and fewer prompts indicating symmetric geometric structures (20%). For a fair comparison, we render 3D assets generated by our method and baselines by circling around the object at a default elevation and camera distance Guo et al. (2023), resulting in 120 images. We then evaluate the gap between the rendered images and reference images generated by Stable Diffusion Rombach et al. (2022) based on the collected prompts. We sample $10k$ points on the generated meshes to calculate 3D metric. To demonstrate that our method is trivially adaptable to various multi-view diffusion models, we randomly use either Zero123 Liu et al. (2023c) or MVDream Shi et al. (2023b) and Zero123++ Shi et al. (2023a) for subsequent experiments. For the effect of different diffusions on the results, please refer to the supplementary for details.

**2D Metrics.** Following DreamFusion Poole et al. (2022), we use CLIP R-score to measure semantic coherence, defined as the probability of rendered images retrieving the correct caption among collected prompts. Additionally, we choose $\text{FID}_{\text{CLIP}}$ Kynkäänniemi et al. (2022) for **image fidelity measurement**, which is calculated by the disparity in distribution between the rendered image and reference image features, both encoded by CLIP ViT-B-32 Radford et al. (2021). We average the metric over 120 rendered images for the quantitative comparison.

**3D Metric.** These metrics mentioned above are for measuring 2D images. Limited by rendering angles and geometric self-occlusion, 2D metrics often struggle to assess 3D objects in 360 degrees fully. To the best of our knowledge, no metrics have yet been introduced in text-to-3D tasks for evaluating the semantic consistency of 3D assets. Therefore, to lift **semantic coherence measurement** from 2D to 3D, we propose using Uni3D Zhou et al. (2024), the largest 3D presentation model with one billion parameters under text-image-pointcloud alignment learning objective. We adopt a similar strategy to the CLIP R-score, except that we replace the image and text encoders in the CLIP with the point cloud and text encoders from the Uni3D, referred to as "$\text{Uni3D}_{\text{score}}$".

**User study.** 3D reconstruction tasks are typically evaluated of the error reconstructed shape compared to the ground truth Ma et al. (2021). However, these metrics are difficult to apply to text-to-3D tasks, as there is no ground truth. We additionally conduct a user study for **geometry consistency measurement**. We collected responses from 20 participants. Each user is presented with a 360-degree perspective of objects and asked to select: whether the 3D object exhibits structural consistency. We then report the rate of consistency as an auxiliary metric, referred to as "Cons. Rate". We collected responses from 30 participants.

**Quantitative Comparison.** In Tab.2, we conduct a quantitative comparison over generation quality, text-image consistency, and 3D consistency. Overall, the results indicate that our method signif-

icantly outperforms the baselines across all metrics, demonstrating that we achieve high-fidelity, text-image, and text-3D consistency in the generated quality while ensuring 3D spatial consistency.

**Qualitative Comparison.** Fig.7 and Fig.5 compare our method with the baselines. In Fig.7, we present four visual examples: the first three rows depict non-symmetric geometries, while the last row is for symmetric geometry. Notably, we display the front, side, and back views, where the back views are highlighted with red rectangles to enhance the observation of potential multiple faces issues. We highlight our improvements in visual comparison in Fig.7. Dreamfusion and ProlificDreame produce high-quality frontal views but fail to form a plausible 3D object. In particular, ProlificDreamer delivers photorealistic 3D assets with semantic coherence, where every view resembles canonical views, i.e., the back views that are shown in red rectangles, are mistakenly optimized as front views, resulting in Janus problems. GSGEN mitigates some of the 3D inconsistencies by introducing 3D priors from the pre-trained Point-E. However, the fidelity of the textures it generates is still insufficient for complete satisfaction. Compared to the three methods mentioned above, MV-Dream stands out as the most effective solution for addressing multi-view inconsistency issues. This is achieved by fine-tuning pre-trained 2D diffusion models using multi-view images rendered from 3D data. Nevertheless, due to the rendering quality and sparsity of 3D training data, the generated results often exhibit cartoon-style textures and semantically lost geometries, particularly when dealing with uncommon and challenging given prompts. For example, it struggles to generate a rocket as required in the second case, the samurai style as required in the third case, and a log as required in the fourth case. By incorporating explicit 3D priors with a 2D diffusion model that is capable of imagination diversity, GeoDream significantly alleviates the multifaceted nature of generated 3D assets, in terms of both meshes and rendered images exhibiting impressive photorealistic textural details, while maintaining semantic faithfulness, as shown in Fig.1 and Fig.7. More analysis and comparisons with other baselines can be found in the supplementary. Finally, we observe that due to the inherent lack of perfect consistency between source views, the constructed cost volume is quite rough as shown in Fig.5. However, the ultimately generated 3D assets tend to produce rich details and more complete and consistent geometry. This suggests that disentangling 3D and 2D priors is a potentially exciting direction, as it provides a flexible way to further refine 3D priors while maintaining the ability of 3D priors to unleash 2D diffusion priors.

### 3.2 ABLATION STUDY

We perform ablation studies to justify the effectiveness of each GeoDream designs. We activate all modules and training strategies mentioned in the Sec.2, except for the modified part described in each ablation experiment below.

**The Effect of 3D Priors.** We visualize the cost volume obtained through the volume construction model, as shown in Fig.4 (a). Fig. 4 (a) combined with (e) demonstrate that relying on rough geometric cues can significantly activate the potential of 3D awareness in 2D diffusion, alleviating the character's tendency to exhibit multifaceted issues. In contrast to fixed priors in Fig.4 (b), we propose using optimizable priors that gradually evolve according to the optimization state, thus producing progressively refined results, as shown in Fig.4 (e) and Fig.4 (j). To further assess its impact, we also attempt to deactivate the cost volume, i.e., randomly initializing the 3D prior. The 3D inconsistency issue also arises, as shown in Fig.4 (f).

**The Effect of Learning Rate Decay Schedule.** We propose to set the learning rate of the cost volume to a smaller value and gradually increase it for geometric detail optimization, aiming to maintain geometric priori cues in the early optimization stage. And vice versa for the learning rate of texture, which can prevent content drift in the later stage of optimization. During the early

Table 2: Quantitative comparison with baselines.

| Model | $\text{FID}_{\text{CLIP}} \downarrow$ | CLIP R-score↑ | | $\text{Uni3D}_{\text{score}} \uparrow$ | Cons. Rate↑ |
| --- | --- | --- | --- | --- | --- |
| | | **B/16** | **L/14** | | |
| DreamFusion Poole et al. (2022) | 59.6 | 0.844 | 0.870 | 0.514 | 0.429 |
| Fantasia3D Chen et al. (2023a) | 49.2 | 0.909 | 0.935 | 0.486 | 0.229 |
| LatentNeRF Metzer et al. (2023) | 58.9 | 0.729 | 0.763 | 0.454 | 0.314 |
| Magic3D Lin et al. (2023) | 58.3 | 0.772 | 0.806 | 0.743 | 0.800 |
| ProlificDreamer Wang et al. (2023d) | 48.8 | 0.866 | 0.892 | 0.629 | 0.257 |
| MVDream Shi et al. (2023b) | 50.6 | 0.852 | 0.886 | 0.771 | 0.829 |
| **Ours** | **47.9** | **0.935** | **0.962** | **0.800** | **0.914** |

optimization stage, we adopt an initially high learning rate to fight early overfitting Li et al. (2019); He et al. (2019). The detailed learning rate curves are depicted in Fig.6. To assess the impact of the learning rate decay schedule, an ablation study is conducted, where the learning rate of the cost volume is set to a suitable constant value. The generated 3D assets still suffer severe degeneration, resulting in a completely collapsed geometry in Fig.4 (c). The reason is that, during the early stage of optimization, there may be a lot of ambiguity and conflict in the appearance information across different views. Hence, during the early optimization stage, we propose to set the learning rate of the cost volume to a smaller value and gradually increase it for geometric detail optimization. And vice versa for the learning rate of texture, which can prevent content drift in the later stage of optimization, please refer to supplementary for details.

We further justify whether we should use texture priors. We report a visual result using a pre-trained texture MLP in Sec.2.1, rather than reinitializing the MLP network and hash encoding in Sec.2.2. Fig.4 (g) shows that introducing texture priors generally leads to a visual appearance that tends toward non-photorealism and over-smoothing. This observation underlines the necessity of introducing only 3D geometric priors, which only contribute to the geometry modeling during the lifting, avoiding compromising the appearance modeling due to texture priors.

**The Effect of Mesh Fine-tuning.** We convert NeuS to DMTet to improve geometric and appearance details. We first show the NeuS-based visual results in Fig.4 (h). GeoDream produces better results with finer details, as evidenced in Fig.4 (j). The reason is that the benefits of the 3D assets we generate, which yield improved 3D consistency, lie in the ability to enhance the accuracy of surfaces, thereby reducing the complexity of texture optimization in the DMTet. Fig.4 (d) presents an ablation study on SDS and VSD loss. SDS is observed to produce over-saturated textures, as opposed to the VSD loss that we default to using.

**The Effect of Rendering Resolution.** Through empirical experimentation, we deduce that collapsed geometry often results in textural distortions, thereby increasing the difficulty of optimization. Hence, we conjecture that 3D consistency is one of the main bottlenecks for increasing the rendering resolution in prior work. Instead, by integrating 3D geometric priors, we achieved better results closer to diffused distributions, making the optimization easier. Consequently, we successfully increase the rendering resolution from 512 to 1024, as shown in Fig.4 (j). Additionally, Fig.4 (i) demonstrates that GeoDream still provides competitive results at $512 \times 512$ resolution.

## 4 CONCLUSION

We significantly improve the rendering fidelity of images and the details of texture meshes, while greatly alleviating the notorious Janus problem. Specifically, our proposed disentangled solution provides geometric cues to the distillation process and allows us to properly utilize the implicit 3D prior present in the large-scale text-to-2D image diffusion models. Additionally, the disentangled design offers a flexible way to optimize 3D priors gradually. The visual and numerical comparisons with the state-of-the-art methods justify our effectiveness and show our superiority over the latest methods in 3D generation.

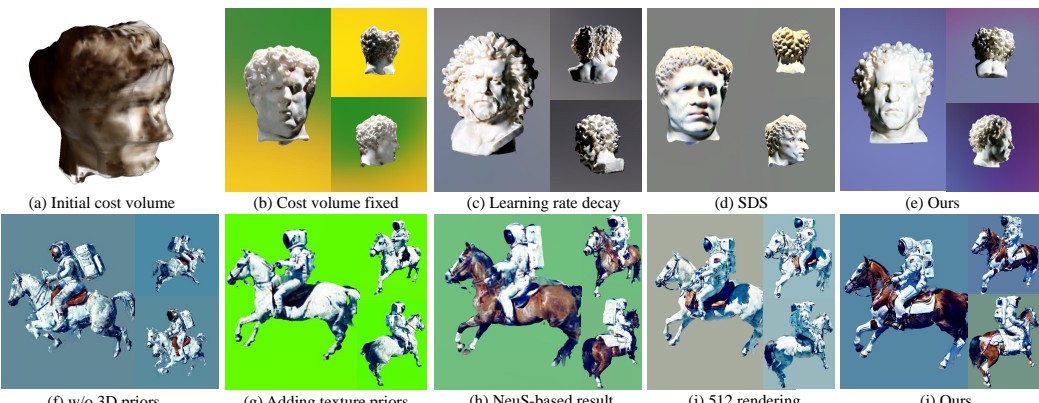

Figure 4: Ablation study of proposed improvements for text-to-3D generation.

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

# A APPENDIX

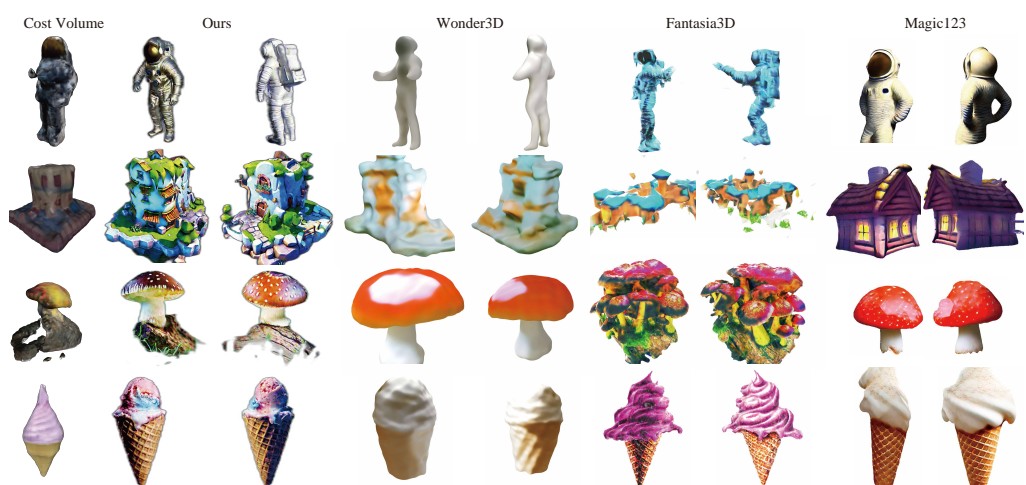

Figure 5: Qualitative comparison with baselines. For each row from up to down, the given prompts are: (1) *3D render of a statue of an astronaut.* (2) *3D stylized game little building.* (3) *A brightly colored mushroom growing on a log.* (4) *An ice-cream cone.*

# B MORE RESULTS

Here, we visualize more qualitative comparison with baselines, as shown in Fig.5 and Fig.7. In contrast to One2345 and MVDream, which only utilize 3D priors trained Objaverse without 2D priors, the insufficiency of high-quality 3D data results in a cartoonish style shift, severely limiting the model's ability to generate 3D assets with semantic consistency. For each row from top to bottom, the given prompts are: (1) Corgi riding a rocket. (2) A brightly colored mushroom growing on a log. (3) Samurai koala bear. MVDream only generates a dog without the rocket and only a mushroom without the log, failing to follow the user's guidelines and creating 3D assets with factual errors. Compared to One2345, our method not only better reflects the semantic consistency but also generates various styles, including photorealistic 3D textures.

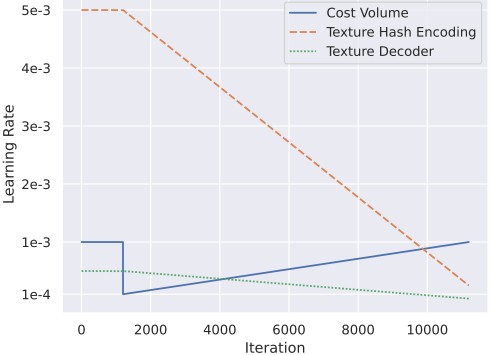

Figure 6: The detailed learning rate schedule.

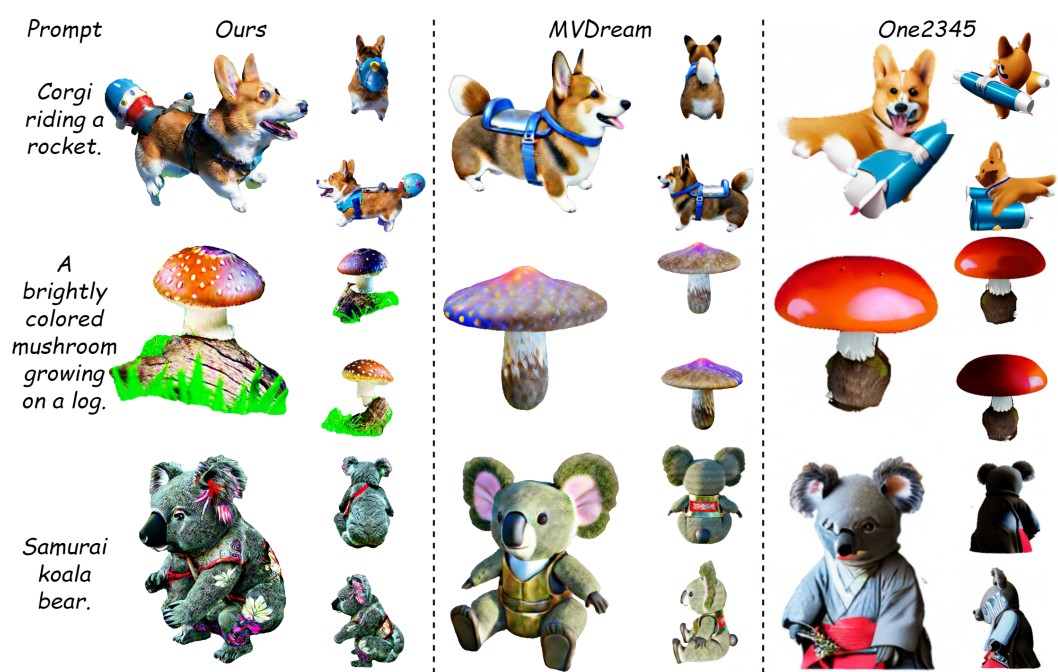

Figure 7: Qualitative comparison with One2345 and MVDream.

## C  VIDEO

Our supplementary material also includes a video, which shows more visualizations and the qualitative comparison with MVDream Shi et al. (2023b), inviting reviewers to watch for a more intuitive visual experience.

## D  SOURCE CODE

To facilitate future research, we will release all the source code and test prompts.

## E  DEFINITION OF THE JANUS PROBLEM

We explain in further detail the definition of the Janus problem (3D inconsistency), which refers to a phenomenon that the learned 3D representation, instead of presenting the 3D desired output, shows multiple canonical views of an object in different directions Wikipedia (2023); Armandpour et al. (2023). For instance, when the given prompt indicates an asymmetric geometric structure, such as a person or an animal, the generated 3D asset has multiple faces but lacks complete and correct back views. In contrast, when the given prompt indicates a symmetric structure, such as a cake or a hamburger, which does not have strictly defined back views, issues of 3D inconsistency typically do not arise. Therefore, when calculating the subjective metric, geometrically symmetric 3D assets do not suffer from 3D inconsistency by default.

## F  VIEWPOINT SAMPLING STRATEGY

We propose a critical viewpoint sampling strategy to enhance the stability of constructing cost volumes. Cost volume-based methods Yao et al. (2018); Zhang et al. (2022); Long et al. (2022); Liu et al. (2023b) rely on the consistency and accuracy of multi-views to find local correspondences and infer geometry. We empirically find that current multi-view diffusion models Liu et al. (2023c); Yang et al. (2023); Liu et al. (2023b); Li et al. (2023c); Shi et al. (2023b); Ye et al. (2023) can

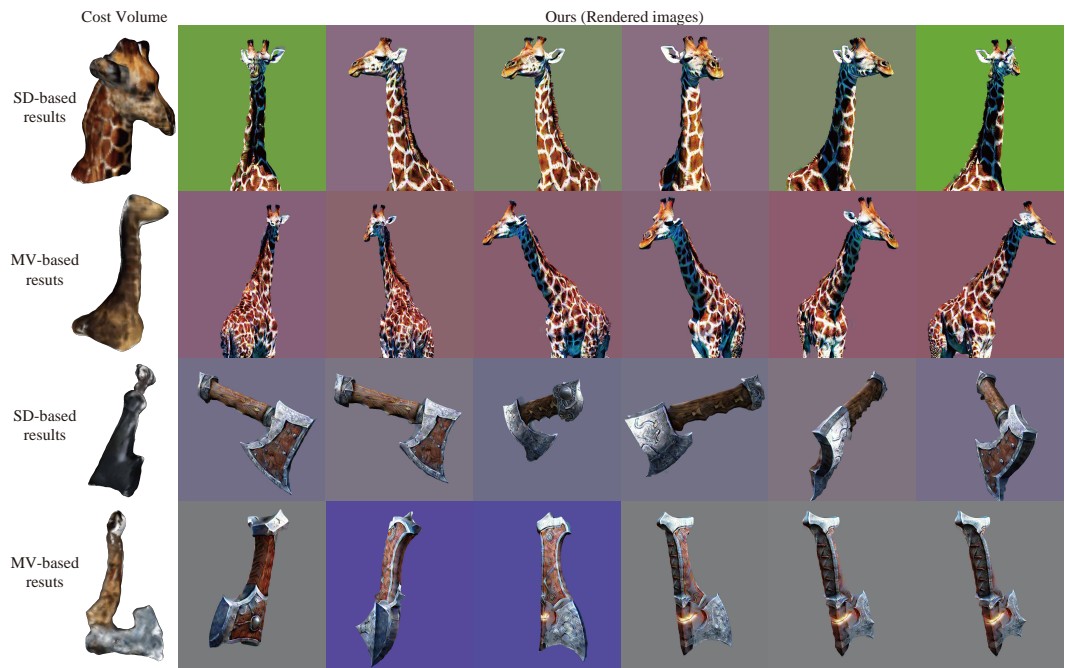

Figure 8: Ablation on the methods for obtaining reference views. We compare the generated 3D assets based on reference views predicted by Stable Diffusion and MVDream, driven by user-provided texts. GeoDream adapt to reference views from various sources. For each row from up to down, the given prompts are: (1) *A majestic giraffe with a long neck.* (2) *Viking axe, fantasy, weapon, blender, 8k, HD.*

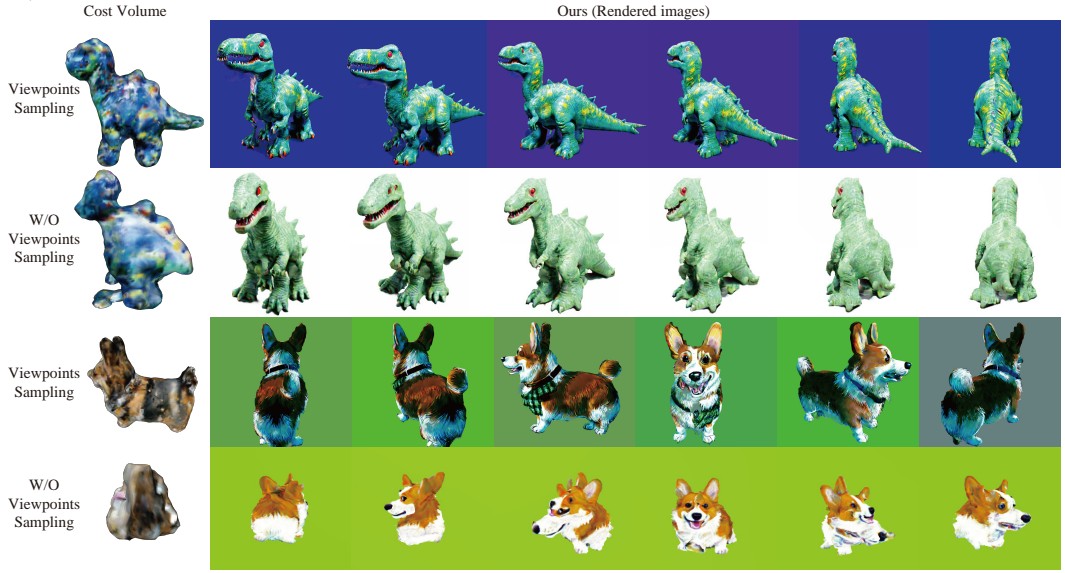

Figure 9: Ablation on the viewpoint sampling strategy. We demonstrate that using our proposed viewpoint sampling strategy contributes to the more robust generation of a consistent cost volume, significantly avoiding the outcomes of geometric collapse. For each row from up to down, the given prompts are: (1) *A dinosaur toy.* (2) *A corgi.*

provide relatively accurate and consistent predictions for the small relative pose when fed with front and side views as reference views. Instead, when a back view is used as the reference view, inconsistencies tend to worsen. Our analysis indicates that these multi-view models are fine-tuned from

2D pre-trained diffusion models, which exhibit weaker performance in predicting non-canonical view information. Additionally, the information implied by back views is quite ambiguous, posing challenges for predicting consistent information. Consequently, we propose a viewpoint sampling strategy to mitigate the aforementioned problems.

Specifically, We obtain reference views driven by a user-provided text in one of two methods: i) Obtaining a front view predicted by Stable Diffusion Rombach et al. (2022), which is trivial as Stable Diffusion often biases towards generating canonical views. ii) Utilizing MVDream Shi et al. (2023b) to output desired views based on our predefined absolute camera positions. In our experiments, following the default settings of MVDream, we set the absolute elevation angle at $15°$ and absolute azimuth angles at $0°$, $90°$, $180°$, and $270°$. We sample four viewpoints on the sphere surface with a default radius to obtain the front, left, back, and right views as reference views.

When the reference view is predicted by Stable Diffusion, we require either Zero123 Liu et al. (2023c) or Zero123++ Shi et al. (2023a) to randomly sample viewpoints within a range of a relative azimuth angle less than $180°$ and a relative elevation angle less than $30°$. Subsequently, we sample an image with a relative azimuth angle of $180°$ and a relative elevation angle of $0°$ to serve as the back view, which is then added to the source views. In the case of reference views predicted by MVDream, we use Zero123 or Zero123++ to sample viewpoints relative to the front view, left side view, and right side views, within a range of a relative azimuth angle less than $45°$ and a relative elevation angle less than $30°$. Subsequently, the back view predicted by MVDream is supplemented by the source views. We show the visualized comparison of the impact of reference views generated by Stable Diffusion and MVDream on the generated 3D assets, as shown in Fig.8. We report visualized results without viewpoint sampling strategy and the results with viewpoint sampling strategy, as shown in Fig.9. The visualized results indicate that our proposed sampling strategy can adapt to reference views predicted by both Stable Diffusion and MVDream, significantly enhancing the quality of the constructed cost volume and the consistency of the generated 3D assets.

Finally, we observe that due to the inherent lack of perfect consistency between source views, the constructed cost volume is quite rough, even with the viewpoint sampling strategy, as shown in Fig.8 and Fig.9. However, the ultimately generated 3D assets tend to produce rich details and more complete and consistent geometry. This suggests that disentangling 3D and 2D priors is a potentially exciting direction, as it provides a flexible way to further refine 3D priors while maintaining the ability of 3D priors to unleash 2D diffusion priors.

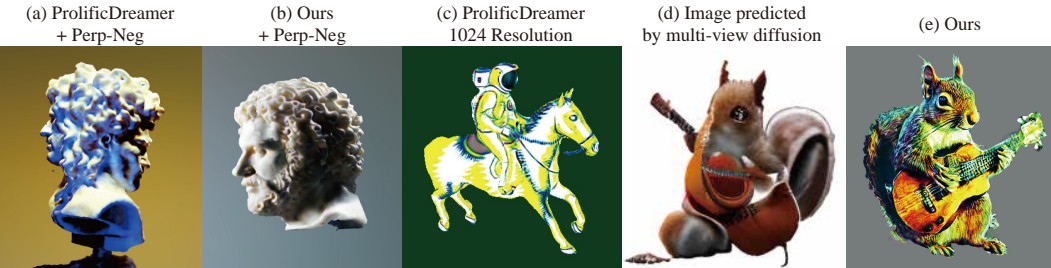

Figure 10: Ablation on negative prompting, rendering resolution, and corner case. The given prompts are: (a) and (b) *A 3D printed white bust of a man with curly hair.* (c) *An astronaut riding a horse.* (d) and (e) *A DSLR photo of a squirrel playing guitar.*

## G  ABLATION ON NEGATIVE PROMPTING, RENDERING RESOLUTION, AND CORNER CASE

**Prompting.** Perp-Neg Armandpour et al. (2023) introduces a negative prompt algorithm that transforms 2D Diffusion into 3D, addressing the Janus problem. We attempt to integrate the negative prompt algorithm into both ProlificDreamer and GeoDream, as shown in Fig.10 (a) and Fig.10 (b). The result shown in Fig.10 (a) demonstrates that the negative prompt algorithm still fails to mitigate the Janus problem stably. Fig.10 (b) illustrates that GeoDream is able to yield consistent 3D assets both with and without the negative prompt algorithm. However, since we did not observe a signif-

icant improvement in the results, we opted not to use the negative prompt algorithm as a default in our experiments. Instead, we employ view-dependent prompting as in previous works Poole et al. (2022); Wang et al. (2023d).

**Rendering Resolution.** We attempt to increase the rendering resolution to 1024 in ProlificDreamer, which typically struggles with over-saturation issues, as demonstrated in Fig.10 (c). Our analysis suggests that the absence of 3D priors often leads to collapsed geometry, resulting in textural distortions and thereby increasing the complexity of the optimization.

**Corner Case.** We further explore the robustness of GeoDream when faced with failures of multi-view diffusion in predicting multiple views. For instance, when the given prompt is "*A DSLR photo of a squirrel playing guitar*", multi-view diffusion struggles to accurately predict the correct spatial relationship between the guitar and the squirrel, due to the sparsity of 3D training data. However, GeoDream excels in preserving the generalizability and creativity of 2D diffusion priors, enabling more effective compatibility with imperfect multi-view predictions, and thus generating semantically correct 3D assets, as shown in Fig.10 (e).

## H TRAINING STABILITY AND DIVERSITY

**Stability.** Prior text-to-3D studies are notoriously brittle. The same hyperparameter settings often lead to vastly different results in terms of complete failure or success, depending on the random seed, making them hard to control. To assess the training stability of GeoDream, we conduct several experiments on the same prompt, as shown in Fig.11. GeoDream exhibits exceptional training stability. The reason lies in the 3D priors we introduced, which significantly reduce the randomness caused by the random seeds.

**Diversity** Additionally, we can generate diverse 3D models by controlling and leveraging the diversity capabilities of Stable Diffusion or MVDream to predict various reference views, as mentioned in Sec.F and Fig.8. In summary, GeoDream provides a balanced solution between diversity and stability.

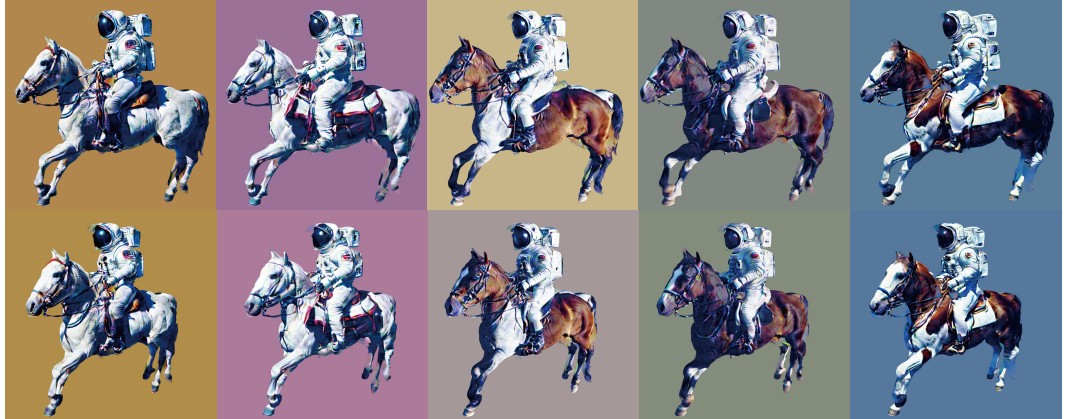

Figure 11: Ablation on training stability. We conduct several experiments on the same prompt to verify the training stability of GeoDream. The given prompt is: *An astronaut riding a horse.*

## I LICENSES

We provide the URL, citations, and licenses of the open-sourced assets we used in this work, as shown in Tab.3.

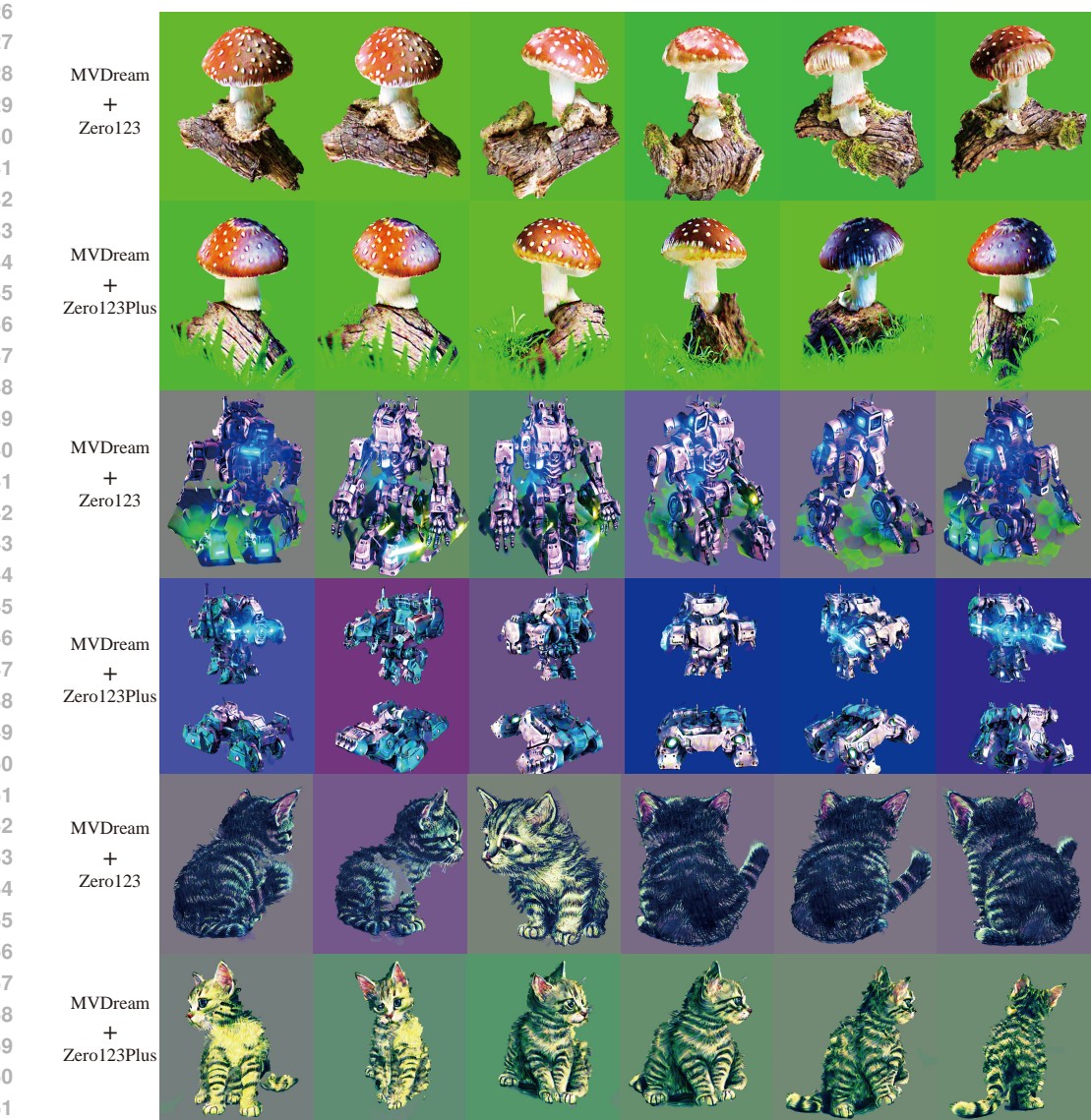

Figure 12: Ablation on source views predicted by different multi-view diffusion models. We compare our generated 3D assets based on source views predicted by Zero123 and Zero123++. For a fair comparison, the reference views are generated by MVDream and driven by user-provided texts. GeoDream adapt to source views predicted by various multi-view diffusion models. For each row from up to down, the given prompts are: (1) *A brightly colored mushroom growing on a log.* (2) *Mech robot with large weapons on top with hexagonal bases.* (3) *A small kitten.*

## J ALGORITHM

We provide a summarized algorithm of priors refinement in Algorithm 1.

## K TRAINING DETAILS

We construct a cost volume with $150 \times 150 \times 150$ voxels in 2 minutes on an NVIDIA-V100-32GB GPU. During the priors refinement stage, we employ a network modified based on ProlificDreamer Wang et al. (2023d). We replace the learnable hash encoding used in ProlificDreamer by cost volume. We choose a single-layer MLP to decode the color from texture hash encoding as

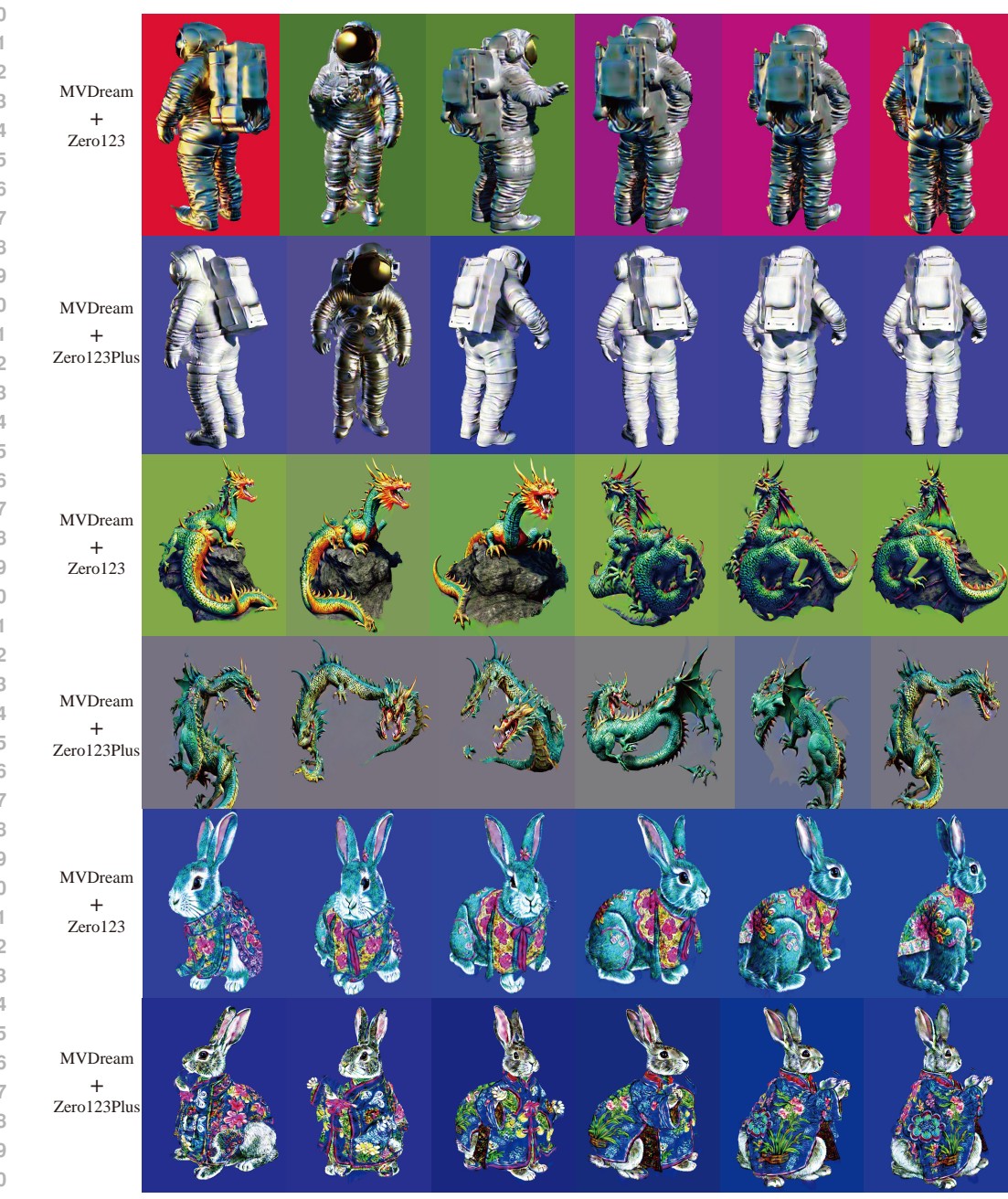

Figure 13: Ablation on source views predicted by different multi-view diffusion models. We compare our generated 3D assets based on source views predicted by Zero123 and Zero123++. For a fair comparison, the reference views are generated by MVDream and driven by user-provided texts. GeoDream adapt to source views predicted by various multi-view diffusion models. For each row from up to down, the given prompts are: (1) *3D render of a statue of an astronaut.* (2) *A high-quality photo of a dragon.* (3) *A cute rabbit in a stunning, detailed Chinese coat.*

Instant-NGP Müller et al. (2022). Following ProlificDreamer, we set the particle to 1 and utilize v-prediction Salimans & Ho (2022) to train the LoRA Hu et al. (2021) based on Stable Diffusion v2.1 model for VSD loss. Notably, even when the rendering resolution increased from 512 to 1024, the training time did not significantly differ from ProlificDreamer. The reason is that 3D assets generated by GeoDream, exhibit fewer artifacts and thus enhanced rendering efficiency. Specifically,

Table 3: URL, citations, and licenses of the open-sourced assets we used in this work.

| URL | Citation | License |
|---|---|---|
| https://github.com/threestudio-project/threestudio | Guo et al. (2023) | Apache License 2.0 |
| https://github.com/bytedance/MVDream | Shi et al. (2023b) | Apache License 2.0 |
| https://github.com/One-2-3-45/One-2-3-45 | Liu et al. (2023b) | Apache License 2.0 |
| https://github.com/cvlab-columbia/zero123 | Liu et al. (2023c) | MIT License |
| https://github.com/SUDO-AI-3D/zero123plus | Shi et al. (2023a) | Apache License 2.0 |
| https://github.com/huggingface/diffusers | Rombach et al. (2022) | Apache License 2.0 |
| https://github.com/allenai/objaverse-xl | Deitke et al. (2023a;b) | Apache License 2.0 |

training the NeuS representation Wang et al. (2021) with the batch size set to 1 typically requires approximately 3 hours on a single NVIDIA-V100-32GB GPU. Mesh finetuning with a batch size of 2 usually requires around 8 hours on a single NVIDIA-V100-32GB GPU. Utilizing larger batch sizes and parallel multi-GPUs training could potentially reduce training times and we leave this exploration in future work.

## L  ABLATION ON SOURCE VIEWS PREDICTED BY DIFFERENT MULTI-VIEW DIFFUSION MODELS

To demonstrate that GeoDream is trivially adaptable to various multi-view diffusion models, we conduct the visual comparison with our generated 3D assets based on either Zero123 or Zero123++. Specifically, for a fair comparison, the reference views are generated by MVDream and driven by user-provided texts. Then, employing the viewpoint sampling strategy proposed in Sec.F, we obtain source views predicted by Zero123 or Zero123++. Fig.12 and Fig.13 show the comparison of our generated 3D assets based on source views predicted by Zero123 and Zero123++. Fig.12 and Fig.13 illustrate that GeoDream can adapt to different multi-view diffusion models, producing 3D assets with plausible geometry and intricate rendering details in visual appearance. The adaptability and seamless integration of GeoDream with various multi-view diffusion models highlight the evolutionary potential of GeoDream, alongside the future advancements of multi-view diffusion models.

---

**Algorithm 1:** Priors Refinement

---

**Input:** A condition $c$, rotation and translation matrix $\{(R_i, T_i)_{i=0}^{N-1}\}$, voxel location $h$, the variance operation $\mathrm{Var}\{\cdot\}$, the projection procedure $P(\cdot, \cdot)$, multi-view diffusion $f_{mv}$, a 2D feature network $f_{2D}$, a 3D feature network $f_{3D}$, a geometric decoder $f_g$, texture decoder $f_t'$, position encoding $E(\cdot)$, 2D diffusion model $\epsilon_{pretrain}$. Learning rate $\eta_1$, $\eta_2$, $\eta_3$,$\eta_4$ and $\eta_5$ for cost volume $V$, hash texture encoding $h_\Omega$, texture decoder $f_t'$, a LoRA diffusion model $\epsilon_l$ and DMTet parameters, respectively.

**1** Initialize 2D feature network $f_{2D}$, 3D feature network $f_{3D}$, and geometry MLP decoder $f_g$ with pretrained parameters obtained from 3D priors training stage. Initialize texture hash encoding and texture decoder $f_t'$ parameterized by $(\theta_2, \theta_3)$. Initialize a LoRA diffusion model parameterized by $l$.

**2 for** *i=0 to N-1* **do**

**3** $\quad$ $F_i^p \leftarrow f_{2D}(f_{mv}(c, R_i, T_i))$

**4** $V_p = f_{3D}(\, \mathrm{Var}\{P(F_i^p, h)\}_{i=0}^{N-1})$

**5** Cost volume $V_p$ parameterized by $\theta_1$.

**6 while** *not converged* **do**

**7** $\quad$ Randomly sample a camera pose $o$. Sample $M$ query points $x_j$ along the view ray based on camera pose $o$.

**8** $\quad$ **for** *j=0 to M-1* **do**

**9** $\quad\quad$ $s_j \leftarrow f_g(E(x_j), V_P(x_j))$

**10** $\quad\quad$ $c_j \leftarrow f_t'(h_\Omega(x_j), x_j)$

**11** $\quad$ $\hat{x} \leftarrow R(\{s_j\}_{j=0}^{M-1}, \{c_j\}_{j=0}^{M-1})$

**12** $\quad$ $\theta_1 \leftarrow \theta_1 - \eta_1 \mathrm{E}_{t,\epsilon,o}[w(t)(\epsilon_{pretrain}(\hat{x}_t, t, c) - \epsilon_l(\hat{x}_t, t, c, o))\frac{\partial \hat{x}}{\partial \theta_1}]$

**13** $\quad$ $\theta_2 \leftarrow \theta_2 - \eta_2 \mathrm{E}_{t,\epsilon,o}[w(t)(\epsilon_{pretrain}(\hat{x}_t, t, c) - \epsilon_l(\hat{x}_t, t, c, o))\frac{\partial \hat{x}}{\partial \theta_2}]$

**14** $\quad$ $\theta_3 \leftarrow \theta_3 - \eta_3 \mathrm{E}_{t,\epsilon,o}[w(t)(\epsilon_{pretrain}(\hat{x}_t, t, c) - \epsilon_l(\hat{x}_t, t, c, o))\frac{\partial \hat{x}}{\partial \theta_3}]$

**15** $\quad$ $l \leftarrow l - \eta_4 \nabla_l \mathrm{E}_{t,\epsilon}||\epsilon_l(\hat{x}_t, t, c, o)) - \epsilon||_2^2$

**16** Mesh fine-tuning, we use DMTet to extract textured mesh from optimized 3D representation parameterized by $(\theta_1, \theta_2, \theta_3)$ and geometry MLP decoder $f_g$. The extracted DMTet parameterized by $\theta_4$. Initialize a LoRA diffusion model parameters $l'$.

**17 while** *not converged* **do**

**18** $\quad$ Randomly sample a camera pose $o$. Render 2D image $\hat{x}$ at pose $o$.

**19** $\quad$ $\theta_5 \leftarrow \theta_5 - \eta_5 \mathrm{E}_{t,\epsilon,o}[w(t)(\epsilon_{pretrain}(\hat{x}_t, t, c) - \epsilon_{l'}(\hat{x}_t, t, c, o))\frac{\partial \hat{x}}{\partial \theta_5}]$

**20** $\quad$ $l' \leftarrow l' - \eta_4 \nabla_{l'} \mathrm{E}_{t,\epsilon}||\epsilon_{l'}(\hat{x}_t, t, c, o)) - \epsilon||_2^2$

**21 return**

---

