# OpenReview forum: "GeoDream: Disentangling 2D and Geometric Priors for High-Fidelity and Consistent 3D Generation"
_ICLR.cc/2025/Conference — Submitted to ICLR 2025_

### Official Review · Reviewer_qJD9 · 2024-10-27

**Soundness:** 3
**Presentation:** 3
**Contribution:** 2
**Rating:** 3
**Confidence:** 4

**Summary:**

This paper addresses the challenge of text-to-3D reconstruction with GeoDream, a method that integrates explicit 3D priors with 2D diffusion priors to capture clear, 3D-consistent geometric structures. The approach uses a multi-view diffusion model to generate posed images, then constructs a cost volume from these images as native 3D geometric priors. The 3D priors are integrated with 2D diffusion priors through a disentangled design. Comparisons with existing methods show that GeoDream generates more 3D-consistent textured meshes with strong semantic coherence.

**Strengths:**

- The method constructs a cost volume as native 3D priors by aggregating predicted multi-view 2D images in 3D space, enhancing multi-view consistency in the generated 3D models.
- By refining geometric priors through a 2D diffusion model, this work shows an improvement in rendering quality and geometric accuracy.
- Comparisons with baselines highlight the method's effectiveness in generating 3D-consistent textured outputs.

**Weaknesses:**

The proposed method design seems incremental compared to the existing work like One-2-3-45 and has several key limitations:

- Generation Quality: The method's quality does not appear to surpass existing approaches. The visual results, even in the teaser of Fig. 1, reveal artifacts around object boundaries and inconsistencies in geometry, especially visible in mismatched normal maps and images. In the supplementary video, examples like the EAGLE, MUSHROOM, and GIRAFFE also exhibit incomplete geometry, likely due to limitations in the generated cost volume. A comparison with RichDreamer (CVPR 2024), which uses a normal-depth diffusion model as a geometry prior, would be valuable.

- Training of 3D Priors: The generalizabilty and quality of the 3D prior rely on the model for multi-view generation and cost-volume construction, raising questions about its generalizability and robustness for diverse prompts. The paper does not describe in detail how the 3D prior model is trained. The tested prompts cover a narrow range, making it difficult to assess performance across broader scenarios.

- Evaluation and Analysis: The evaluation could be stronger. For example, the user study had only 20 to 30 participants, which may be too small for reliable comparisons. Additionally, the ablation study lacks quantitative metrics to substantiate the design choices.

- Optimization Time: The method requires a significant optimization time, taking around 8 hours per instance, which limits its practicality for broader applications.

**Questions:**

- Clarification of the novelty.
- clarification of  the sources of artifacts and inaccuracies in the generated geometry? These issues are visible in Fig. 1 and more pronounced in the video examples.
- How does GeoDream compare to RichDreamer (CVPR 2024) in terms of geometric detail and model accuracy?
- Could the authors provide a more thorough user study and ablation study to validate their approach across a broader range of metrics and prompts?

---

### Official Review · Reviewer_ZSUz · 2024-10-28

**Soundness:** 2
**Presentation:** 3
**Contribution:** 2
**Rating:** 3
**Confidence:** 4

**Summary:**

The paper tackles the task of Text-to-3D generation, addressing the technical challenge of the Janus problem that previous methods have encountered. The key insight and motivation is to leverage 2D diffusion priors to enhance the generation quality. The approach involves using a multi-view diffusion model to generate multi-view images, representing the 3D object with a cost volume, and then optimizing the 3D object using differentiable rendering from the generated multi-view images. In the second stage, the geometry decoder is fixed, and the texture decoder is optimized based on the SDS loss, resulting in improved 3D object generation from textual descriptions.

**Strengths:**

1. The ablation studies are sufficient to validate the effectiveness of the proposed components.
2. The paper is well written and easy to read.

**Weaknesses:**

1. The paper only compares with works from 2023 and does not include comparisons with more recent papers from CVPR 2024 and ECCV 2024. I have observed that some papers have better visualization results than this paper, such as One-2-3-45++. It would be beneficial for the paper to include these comparisons to demonstrate its standing in the current research landscape.
2. I do not understand the rationale behind the second phase, which is the optimization of the texture decoder using the SDS loss. In my view, if the multi-view diffusion model performs well, the SDS loss would be unnecessary, as evidenced by some recent papers. Is it because the multi-view diffusion model trained in this paper does not generate satisfactory results?
3. The use of cost volume to define the 3D representation seems odd. While I understand that cost volume adds 3D regularization, it has several drawbacks, including: (a) The cost volume essentially fuses multi-view features. If the input images have occlusion relationships, the cost volume cannot fuse to obtain the correct feature volume. (b) The spatial complexity of cost volume is quite large, which limits the resolution.
4. Overall, the paper's design shows some technical improvements compared to 2023's work, but it seems somewhat outdated compared to the work presented at CVPR 2024 and ECCV 2024. Although the ablation studies are complete, I still do not believe this paper is suitable for acceptance by ICLR.
5. The paper should present more generation results from the multi-view diffusion model to demonstrate its effectiveness, as this model is the core technology of the paper. It is also necessary to clarify how this diffusion model improves upon MVDream. Personally, I do not believe that improvements in 3D representation or training methods can bring about fundamental enhancements. The essence of 3D generation is to learn a stronger 3D prior, which may be represented by a multi-view diffusion model or a direct 3D generative model.

**Questions:**

Authors should compare with SOTA baseline methods and demonstrate the advance of method design.

---

> ### Comment · Reviewer_ZSUz · 2024-11-26
>
> Since there is no response from authors, I will maintain my score.

---

### Official Review · Reviewer_EBL4 · 2024-11-04

**Soundness:** 3
**Presentation:** 3
**Contribution:** 2
**Rating:** 6
**Confidence:** 4

**Summary:**

This paper disentangles 2D and 3D priors on text-to-3D generation and can generation up to 1024 resolution.

**Strengths:**

1. Higher resolution text-to-3D generation
2. Disentangled 2D and 3D representation
3. Results are good
4. They combined Neus and DMTet as 3D representation

**Weaknesses:**

1. There are still some artifacts on the extracted mesh.

**Questions:**

It would be better if the authors can show comparison with baseline methods in the videos.

---

### Official Review · Reviewer_xGa4 · 2024-11-04

**Soundness:** 3
**Presentation:** 3
**Contribution:** 2
**Rating:** 3
**Confidence:** 4

**Summary:**

GeoDream presents an innovative approach to 3D asset generation by integrating disentangled 3D priors with 2D diffusion models. This method leverages a multi-view diffusion model to generate posed images, subsequently constructing a cost volume as a 3D prior to ensure spatial consistency. By separating 2D and 3D priors, GeoDream allows iterative refinement of these priors, enhancing the model’s 3D awareness without compromising the diversity or fidelity of generated 3D models.

**Strengths:**

1. The disentangled approach to handling 2D and 3D priors effectively addresses the Janus problem, resulting in improved 3D consistency.
2. The framework supports rendering up to 1024x1024, surpassing most SDS-based methods in resolution.
3. GeoDream demonstrates robustness across a wide range of prompts and asymmetric structures.

**Weaknesses:**

1. Lack of comparative analysis with state-of-the-art methods such as LucidDreamer (CVPR 2024), RichDreamer (CVPR 2024), and ImageDream (Arxiv 2023).
2. While primarily compared with SDS-based methods, GeoDream only includes Wonder3D as an exception. Given the limitations of SDS in terms of low-quality mesh and texture, comparisons with direct reconstruction methods such as InstantMesh (Arxiv 2024), One-2-3-45++ (CVPR 2024), and Era3D (NeurIPS 2024) would strengthen the evaluation.
3. The provided video reveals low mesh quality in certain outputs.
4. Additional pure mesh comparisons with other methods are needed to better assess mesh quality.


Reference:

[1] Liang, Yixun, et al. Luciddreamer: Towards high-fidelity text-to-3d generation via interval score matching. CVPR. 2024.

[2] Qiu, Lingteng, et al. Richdreamer: A generalizable normal-depth diffusion model for detail richness in text-to-3d. CVPR. 2024.

[3] Wang, Peng, and Yichun Shi. Imagedream: Image-prompt multi-view diffusion for 3d generation. arXiv. 2023.

[4] Xu, Jiale, et al. Instantmesh: Efficient 3d mesh generation from a single image with sparse-view large reconstruction models. arXiv. 2024.

[5] Liu, Minghua, et al. "One-2-3-45++: Fast single image to 3d objects with consistent multi-view generation and 3d diffusion. CVPR. 2024.

[6] Li, Peng, et al. Era3D: High-Resolution Multiview Diffusion using Efficient Row-wise Attention. NeurIPS 2024.

**Questions:**

1. Table 2 appears to have a formatting issue—could you clarify the template used?
2. Why does Figure 3 present the generated mesh using rendered RGB rather than pure mesh? This choice limits the ability to assess mesh quality directly.

Please see more questions in the weaknesses section.

---

### Comment · Area_Chair_aQWZ · 2024-11-21
**Please initiate discussions!**

Dear authors and reviewers,

The discussion phase has already started. You are highly encouraged to engage in interactive discussions (instead of a single-sided rebuttal) before November 26. Please exchange your thoughts on the submission and reviews at your earliest convenience.

Thank you,
ICLR 2025 AC

---

### Comment · Area_Chair_aQWZ · 2024-11-25
**Last day for interactive discussions!**

Dear authors and reviewers,

The interactive discussion phase will end in one day (November 26). Please read the authors' responses and the reviewers' feedback carefully and exchange your thoughts at your earliest convenience. This would be your last chance to be able to clarify any potential confusion.

Thank you,
ICLR 2025 AC

---

### Meta-Review · Area_Chair_aQWZ · 2024-12-19

**Metareview:**

The submission received negative reviews from the reviewers. The reviewers' main concerns were mostly around the quality of the results and insufficient analysis. The authors did not submit a rebuttal. After reading the paper and the reviewers' comments, the AC agrees with the decision by the reviewers and recommends rejection.

**Additional Comments On Reviewer Discussion:**

The reviewers raised several questions regarding lack of comparisons with state-of-the-art methods (xGa4, ZSUz, qJD9), low quality in the results (xGa4, qJD9), and design choices (ZSUz). The AC ignores the review from EBL4 due to lack of review quality. The authors did not provide a response.

---

### Decision · Program_Chairs · 2025-01-22

Reject